# What Is the Current Direction of the Research on Carotenoids and Human Health? An Overview of Registered Clinical Trials

**DOI:** 10.3390/nu14061191

**Published:** 2022-03-11

**Authors:** Daniela Martini, Letizia Negrini, Mirko Marino, Patrizia Riso, Cristian Del Bo, Marisa Porrini

**Affiliations:** 1Department of Food, Environmental and Nutritional Sciences (DeFENS), Università degli Studi di Milano, 20133 Milano, Italy; daniela.martini@unimi.it (D.M.); letizia.negrini@studenti.unimi.it (L.N.); mirko.marino@unimi.it (M.M.); patrizia.riso@unimi.it (P.R.); marisa.porrini@unimi.it (M.P.); 2CRC Innovation for Well-Being and Environment (I-WE), Università degli Studi di Milano, 20122 Milano, Italy

**Keywords:** carotenoids, bioactive compounds, bioavailability, human study, foods, supplements

## Abstract

Carotenoids have been the object of numerous observational, pre-clinical and interventional studies focused on elucidating their potential impacts on human health. However, the large heterogeneity among the trials, in terms of study duration and characteristics of participants, makes any conclusion difficult to draw. The present study aimed to explore the current carotenoid research trends by analyzing the characteristics of the registered clinical trials. A total of 193 registered trials on ClinicalTrials.gov and ISRCTN were included in the revision. Eighty-three studies were performed with foods, one-hundred-five with food supplements, and five with both. Among the foods tested, tomatoes and tomato-based foods, and eggs were the most studied. Lutein, lycopene, and astaxanthin were the most carotenoids investigated. Regarding the goals, 52 trials were focused on studying carotenoids’ bioavailability, and 140 studies investigated the effects of carotenoids on human health. The main topics included eye and cardiovascular health. Recently, the research has focused also on two new topics: cognitive function and carotenoid–gut microbiota interactions. However, the current research on carotenoids is still mostly focused on the bioavailability and metabolism of carotenoids from foods and food supplements. Within this context, the impacts/contributions of food technologies and the development of new carotenoid formulations are discussed. In addition, the research is still corroborating the previous findings on vision and cardiovascular health. Much attention has also been devoted to new research areas, such as the carotenoid–microbiota interactions, which could contribute to explaining the metabolism and the health effects of carotenoids; and the relation between carotenoids and cognitive function. However, for these topics the research is still only beginning, and further studies are need.

## 1. Introduction

Carotenoids are a category of bioactive compounds widely found, not only in the plant kingdom (e.g., fruits, vegetable, algae, and fungi), but also in some animal products (e.g., eggs and fish). They form a group of more than 700 fat-soluble compounds, which generally contribute to the yellow to red colors of many foods, although colorless carotenoids such as phytoene and phytofluene have also been identified [1]. The main carotenoids found in the plasma of human subjects consuming carotenoid-rich foods include lycopene, α and ß-carotene, lutein, zeaxanthin, and ß-cryptoxanthin, though the intake of specific sources (e.g., astaxanthin rich fish) can also provide other compounds. These molecules are divided into hydrocarbon carotenes, i.e., containing only carbon and hydrogen (e.g., α- and ß-carotene), and xanthophylls (e.g., lutein) characterized by the presence of oxygen atoms as well, which makes them less hydrophobic [2]. The content of carotenoids in foods is variable and dependent on several factors, including type of food, maturity stage, variety, seasonality, climatic conditions, processing, and storage conditions. Commonly, carotenoids are quite stable regardless of cooking and storage, but exposure to extremely high temperatures can alter them and induce destruction. Numerous studies and systematic reviews have been performed to analyze the determinants of carotenoid bioavailability, concluding that the efficiency of their absorption can be affected by numerous intrinsic and extrinsic factors [3,4,5]. Intrinsic factors: dietary factors, including characteristics of the food matrix that could reduce their bioaccessibility; food processing is able to increase and/or reduce carotenoids’ bioaccessibility, bioavailability, and degradation; the presence of fat and fat-soluble micronutrients that could improve their absorption [6,7]. Extrinsic factors: host-related factors such as presence of diseases, life-style habits, gender, age, and genetic polymorphisms, which could positively or negatively compromise their absorption and metabolism [8,9,10].

Dietary intake and blood carotenoid concentrations widely vary among populations. For instance, the total intake of carotenoids varies from 1 to 22 mg/day (serum/plasma concentration range 1–2.2 µmol/L) in European countries, and from 5 to 16 mg/day (serum/plasma concentration range 1.2–2.5 µmol/L) in the United States [11]. However, carotenoids also vary within the same population, depending on the availability of food sources; and individual factors such as age, sex, health status, and genetic factors [11]. Others important aspects include the methods used for the estimation of carotenoid intake (e.g., food frequency questionnaire, 24-h recall, or food diary), the tools available (e.g., food databases) and the analytical methodologies used for the quantification of biological samples. In addition, compared to other bioactives (e.g., glucosinolates and polyphenols) that are rapidly metabolized and excreted, carotenoids have shown longer half-lives; thus, they may represent a good indicator of a person’s dietary habits.

In the recent COST ACTION CA15136 “EUROCAROTEN,” researchers have been exploring different topics related to carotenoids that could lead to new technologies and/or high-quality foods. Overall, they aim to increase the knowledge, research, and competitiveness of the European agro-food industry [12,13,14,15,16]. An additional important goal the EUROCAROTEN network is trying to promote is the enhancement of knowledge on the health-related effects of these compounds by collecting data and coordinating studies that will reveal the mechanisms of health impacts [17], provide evidence on health biomarkers [18], and indicate health-promoting nutritional recommendations [11].

Many biological functions have been attributed to carotenoids, such as forming a precursor of provitamin A, being activators of nuclear hormone receptors and the immune response, acting as signaling molecules, and acting as antioxidant and anti-inflammatory agents. Furthermore, a large body of epidemiological observations has documented that dietary intake of carotenoids is inversely correlated with obesity, retinopathy, and cataracts; and even severe diseases, such as cardiovascular diseases, diabetes, and some types of cancer [19]. Such protection against chronic diseases has been documented for total blood carotenoid concentration > 1 µM [11]. These promising results were the basis for designing and performing human intervention trials focused not only on the study of carotenoids’ bioavailability from different food sources, but also on corroborating the evidence for their potential health benefits. In fact, numerous recent systematic reviews of human intervention studies concluded that, despite being consistent and promising, the current evidence is still insufficient to substantiate a beneficial effect due to the high number of studies showing mixed and otherwise questionable results [20,21,22,23]. Thus, further studies are needed to provide a comprehensive understanding of the mechanisms through which carotenoids may exert their health effects, and their clinical relevance should be determined in clinical studies.

The present systematic review explores the current trends in the clinical research on carotenoids and gives an overview of the evolution of efforts on this topic in the last 20 years. We also provide useful information for designing and performing future studies. While we have recently published analogous analyses on polyphenols [24] and glucosinolates [25], to the best of our knowledge, no reviews are available on registered clinical trials aimed at evaluating the relationships between carotenoids, and bioavailability and human health.

## 2. Materials and Methods

### 2.1. Database Search Strategy

The present review was conducted using two different registers: “ClinicalTrials.gov” and International Standard Randomized Controlled Trial Number (ISRCTN). The focus was on registered clinical trials investigating carotenoids and performed from 2000 to October 2021. The search was conducted on 1 July 2021 and updated on 4 October 2021 to identify additional studies. The search was performed using the following syntax: carotenoids OR carotene OR lycopene OR xanthophyll OR lutein OR zeaxanthin OR cryptoxanthin OR canthaxanthin OR astaxanthin.

### 2.2. Study Selection

Inclusion required the use of one or more carotenoid in a human intervention study investigating its bioavailability and/or effect on one or more marker of human health. Carotenoids could be provided through foods in which they are naturally present or fortified foods, or as supplements containing only these bioactive compounds, administered as single compounds or as combinations of different carotenoids.

To avoid confounding factors, the only exclusion criterion adopted was supplementation with other dietary bioactives or drugs. No restrictions on the characteristics of the participants or other specific restrictions were applied.

The detailed list of eligibility criteria, developed by following the population, intervention, comparison, outcome, study design (PICOS) format, is reported in Table 1.

### 2.3. Data Extraction and Analysis

Data extraction from the two registers was performed by two reviewers (D.M. and L.N.), and a third author (C.D.B.) checked the extracted information and solved disagreements. For each study, the following information was collected: registration number, registration year, country, characteristics of participants (i.e., sex, age, and health status), study design, type of intervention, primary outcome, and other outcome measures. Studies were classified into four main categories based on their start dates (2000–2004, 2005–2009, 2010–2014, and 2015–2021), similarly to previous studies [24,25]. Within these different time categories, studies were then further classified as studies on foods (naturally rich or fortified) and supplements (i.e., carotenoid-rich extracts or single compounds). Based on the countries in which the studies were performed, locations were classified as “low” (number of registered studies in that country lower than 10), “medium” (10 to 49), and “high” (50+).

## 3. Results

### 3.1. Selected Clinical Trials

The complete flow diagram is shown in Figure 1. A total of 531 registered clinical trials were identified from ClinicalTrials.gov and ISRCTN registries. Among them, no duplicates were found; thus, all trials were assessed for eligibility. Out of these, 173 records were removed because they were not relevant and 165 because of not matching the inclusion criteria. As a result, a total of 193 registered clinical trials were included in the final evaluation. In detail, 83 trials were performed with foods, 105 with food supplements, and 5 with both.

The main characteristics of the 193 selected studies are reported in Table 2. In addition, the full list of the trials included is provided as Appendix A. Out of the 193 trials, 158 were chronic interventions, 33 were acute studies, and only 2 performed both chronic and acute interventions. Most of the trials were randomized (~88%). Regarding blindness, the trials using food were mostly open label (~46%); the others were single blind (~21%), double blind (about 17%), triple blind, or quadruple blind (both 6%).

Conversely, trials using supplements were mostly conducted quadruple blind (around 34%), followed by double blind (~27%), open label (14%), triple blind (12%), and single blind (6%).

Regarding the study design, most of the registered trials were conducted in parallel (52%). About 25% used a crossover design, and the remaining ~22% used a single-arm design, or did not provide information (NA) about the study design. In trials using food, parallel studies accounted for ~45%, crossover studies for 40%, and single group assignment or NA for 16%. In trials using supplements, the most represented study design was parallel (59%), followed by single group assignment or NA (28%), and finally crossover (16%). Regarding location, most of the studies (about 38%) were performed in the United States, followed by Israel (7%) and the United Kingdom (6%).

Using the different time periods, we can see an increasing trend from 2000 to the present of trials performed using foods and supplements (Figure 2). The highest number of such studies was registered in the 2015–2021 period. In detail, there were more studies conducted with the use of supplements in 2000–2004 and 2005–2009, compared to those registered in 2010–2014, when the number of studies based on the use of food supplements was almost equal to those performed on foods.

Only five trials have evaluated both supplements and foods since 2000, one of which was conducted in the first time period (2000–2004). Two were conducted in 2005–2009, and the last two in 2015–2021.

### 3.2. Trials on Carotenoid-Rich Foods

Among the registered trials on carotenoid-rich foods, tomatoes and tomato-based foods (e.g., tomato juice, tomato sauce, or tomato extracts) were the most studied (Figure 3a), with a total of 20 trials, which were performed most frequently in the most recent time period (Figure 3b). Other foods commonly evaluated included eggs and maize (11 and 8 trials, respectively), followed by green vegetables (*n* = 6) and mango (*n* = 4). All these studies were mainly registered during 2015–2021. Other less studied food categories were sweet orange, oils, salmon, rice, raspberries, watermelon, kiwifruit, and pink grapefruit.

### 3.3. Trials on Supplements

The most widely used supplements of carotenoids are reported in Figure 4a. Among the 103 registered trials on supplements, about 41% used ocular pigment carotenoids (i.e., lutein, meso-zeaxanthin), which saw an increase from 4 studies in the 2000–2004 period to 16 trials in the 2015–2021 period (Figure 4b). Lycopene was found in 27 studies. Beta-carotene was used as supplement or administered by the alga *Dunaliella bardawil*, which is particularly rich in 9-cis-beta-carotene.

### 3.4. Characteristics of Subjects Recruited in the Trials

The information on the health status of the subjects recruited in the registered trials is reported in Figure 5a. Most of the trials were conducted with healthy subjects (*n* = 112 trials, 58%), followed by subjects with diseases (*n* = 68 trials, 35%), including eye diseases (e.g., age-related macular degeneration (AMD), glaucoma, night blindness, and *retinitis pigmentosa*), vascular diseases, hypercholesterolemia, coronary heart disease (CHD), atherosclerosis, diabetes, metabolic syndrome, prostate cancer, and other disorders. Finally, some subjects chosen based on risk factors (*n* = 13 studies, 7%) for CVDs, prostate cancer, or vision-related diseases (retinopathy in premature infants and macular degeneration).

Regarding age, 93 registered clinical trials were conducted on both adults and older subjects. Seventy-five were conducted on adults only; 13 on children; 3 on children and adults; and 9 on children, adults, and older subjects (Figure 5b). Finally, as regards sex, 25 studies were conducted with females only and 28 with males only; 139 trials were conducted with participants of both sexes (Figure 5c).

### 3.5. Main Goals of the Registered Trials

Figure 6a depicts the main goals of the registered trials. Out of the 193 trials considered, 140 studies investigated the effects of carotenoids on human health (49 from food, 89 from supplements, and 2 from both), 52 focused on bioavailability (33 from food, 16 from supplements, and 3 from both), and only 1 investigated both health effects and bioavailability. Figure 6b shows that the number of dietary intervention studies assessing health effects has increased over time, from 13 trials in 2000–2004 to 56 in the most recent period (2015–2021). Regarding bioavailability, eight trials were registered in 2000–2004. There was a major increase in the years 2010–2014 (*n* = 20), and then a slight decrease in the last 6 years (*n* = 17).

### 3.6. Main Outcomes of the Registered Trials

The outcomes of the registered trials are shown in Figure 7a. The most frequently evaluated outcomes were the concentrations of carotenoids, or their metabolites, at the plasma level (69 trials). The number of trials doing so increased from 10 studies in 2000–2004, to 25 trials in 2015–2021 (Figure 7b). A total of 43 registered trials considered eye-related outcomes, such as the density of the optic macular pigment, visual acuity, and contrast sensitivity. The number of studies on these outcomes increased over time, from 2 in 2000–2004 to 18 trials in 2015–2021.

The concentrations of carotenoids, or their metabolites, in breast milk, prostate tissue, or skin, were the main outcomes in 16 trials, mostly of which (*n* = 7) were registered in 2015–2021. Additionally, endothelial/vascular function was investigated in a total of 16 trials, followed by oxidative stress and antioxidant status (13 trials), lipid profile and blood pressure (10 trials), and cognitive aspects (9 trials).

## 4. Discussion

To the best of our knowledge, this is the first review to summarize the characteristics of registered clinical trials on carotenoids and carotenoid-rich foods. The current situation shows the renewed interest in these bioactives, as evidenced by the high number (*n* = 193) of clinical trials registered during 2000–2021. Most of the trials were conducted on supplements (*n* = 105), followed by those on carotenoid-rich foods (*n* = 83), reflecting the growing interest of the food and nutraceutical industries. (Only a few used and mixture of supplements and foods (*n* = 5)).

The carotenoids market has been valued to grow up to USD 2.0 billion by 2026 [26]. This is attributed to the increasing use of carotenoids as food colorants and innovations in the technology used for their extraction and synthesis. Carotenoids are also widely used by cosmetics and animal feed industries, and since they have putative health effects as nutraceuticals. However, their potential health effects cannot be definitively claimed without demonstrations of their bioavailability.

In the present revision, 52 trials (33 on foods, 16 on supplements, and 3 on food and supplements) were focused on carotenoid bioavailability. The major food sources analyzed included orange juice, maize, golden rice, mango, peppers, and eggs. In addition, some studies evaluated the impacts of food processing (lutein from fresh and dried beverages), food processing and mastication (lycopene from tomato), and biofortification (ß-cryptoxanthin from maize). Regarding supplements, bioavailability has been investigated mainly for astaxanthin (for which less other information is available) followed by lutein, lycopene, and beta-carotene. Specifically, some studies were devoted to understanding the metabolism of ß-carotene and lycopene, others to studying the influence of proteins and/or new formulations; and others to determining host-related factors affecting the absorption and serum levels of lycopene.

Regarding the impacts of carotenoids and carotenoid-rich foods on markers of human health, most of the registered trials were performed with food supplements (*n* = 69), followed by foods (*n* = 49), and finally, both (*n* = 2). Among dietary supplements, lutein, lycopene, and astaxanthin were the principal compounds investigated. The main topics of research included the prevention and/or treatment of eye diseases and risk reduction of diabetes and CVDs. Most of the studies were medium–long term and performed in adults and older subjects. With respect to eye diseases, the majority of the registered clinical trials tested the effects of lutein alone or in combination with zeaxanthin and mesoxanthin in the treatment of macular pigment optical density, visual acuity, central macular thickness, glaucoma, and *retinitis pigmentosa*—mainly in diabetic subjects (other subjects had relevant vision problems). The interventions varied in duration (depending on the outcome measured) between 4 and 12 months. Only a few studies were carried out for long periods (maximum 48 months). The doses of carotenoids administered within the studies ranged from 1 to 30 mg, depending on time and compound tested (e.g., lutein: 4–20 mg; zeaxanthin: 1–20 mg; mesoxanthin: 12 mg). The relationship between carotenoids and vision has always been a source of discussion among scientists due to the different results obtained [27,28,29]. For example, the Lutein Antioxidant Supplementation Trial demonstrated a significant beneficial effect of lutein supplementation (10 mg/d per 1 year), either alone or in combination with other antioxidants, in subjects with atrophic age-related macular degeneration (AMD) [30]. The Carotenoids and Antioxidants in Age-Related Maculopathy Italian Study showed that lutein supplementation (10 mg/d per 1 year) attenuated dysfunction in the central retina, improved visual acuity, and reduced glare sensitivity, in people with non-advanced AMD [31]. Conversely, the Age-Related Eye Disease Study 2 failed to prove the efficacy of supplementation with lutein (10 mg/d per 1 year) and other micronutrients (e.g., zinc, vitamin C, and E) in terms of reducing the progression of advanced AMD or improving visual acuity [32]. We suggest further studies, including long-term clinical trials (more than 1 year) on large cohorts of individuals, in which the optimal dosages and the routes of administration of carotenoids should be identified in view of their use for the prevention or treatment of age-related eye diseases.

Concerning the roles of carotenoids in diabetes and CVD risk, we have found that lycopene, lutein, astaxanthin, and ß-carotene are the main carotenoids that have demonstrated modulatory protective effects. The majority of relevant studies were performed with subjects with CVD risk factors, but some used healthy subjects. The main markers analyzed were those related to diabetes (e.g., glycaemia and insulin response), lipid profile (in particular, LDL:cholesterol), markers of vascular health (e.g., vascular cell adhesion molecules and vascular/endothelial reactivity), and oxidative stress (e.g., markers of lipid peroxidation and endogenous enzymatic activity). The interventions lasted 1–24 weeks. The concentrations were: lycopene, 8–30 mg; lutein, about 20 mg; astaxanthin, 12–16 mg; and ß-carotene, 6–8 mg.

The role of carotenoids in cardiovascular health has been largely debated. Numerous observational trials have found inverse associations between plasma carotenoid concentrations and risk for CVD and CVD mortality [33,34,35,36]. A recent systematic review has summarized the evidence derived from randomized controlled trials on the effects of carotenoids on CVD health outcomes (e.g., oxidative stress markers and lipid profile) [34]. The results of the review suggest an overall positive contribution of carotenoids to cardiovascular health; however, the results were not always consistent due to the types and dosages of carotenoids, the lengths of the intervention periods, the markers analyzed, and the characteristics of the subjects. The effects seem to be more pronounced in subjects with metabolic syndrome, those with greater oxidative stress and inflammation, and those with impaired vascular function.

About carotenoid-rich foods, most studies were focused on tomatoes and tomato-based products, eggs, and green vegetables. Other foods included maize and mango. Tomatoes and tomato-based products were studied mainly for their positive impacts on markers of inflammation (pro and anti-inflammatory cytokines) and vascular function (vascular, platelet function). The foods tested included tomatoes, lycopene-enriched tomato juice, tomato sauce, tomato ketchup, and tomato extracts. Studies included healthy adults and older individuals, apart from one study carried out on children/adolescents with liver steatosis. The amount of tomato products was variable (e.g., from 80 g tomato sauce up to 300 g raw tomatoes; amount of lycopene rarely provided), as was the duration of the interventions (from 7 days up to 16 weeks). The health effects of tomatoes and tomato-based products have been evaluated in numerous dietary intervention studies in our laboratory: we showed their capacity to counteract oxidative stress and inflammation in healthy individuals [37,38], in line with other studies [21,39]. For example, in a meta-analysis of 17 studies, Cheng and colleagues [21] reported that tomato consumption was associated with significant reductions in LDL:cholesterol and inflammatory markers (IL-6), and an improvement in vascular function. Li and coworkers [39] documented in a meta-analysis of six randomized controlled trials reductions in total and LDL cholesterol, and triglycerides; and an increase in HDL, following tomato consumption. In addition, reductions in the levels of oxidative stress markers, such as lipid and DNA damage, were also reported. Conversely, Tierney et al. [33] concluded in their meta-analysis of 43 human intervention studies that findings for lycopene-rich foods in terms of blood pressure and lipid profile are conflicting. To the best of our knowledge, specific registered trials on this topic are not ongoing; however, blood pressure and serum lipid profile are included as secondary outcomes in numerous studies.

Eggs (enriched with lutein or zeaxanthin; amount not reported) were investigated in the context of eye health (retina function, macular pigments) and cognitive function (e.g., measure of attention, executive function) in both adults and in older subjects, and recently also in adolescents. The trials included the consumption of two eggs per day for a period of 6–24 weeks. The effect of egg consumption on vision has been poorly investigated. The results of a recent meta-analysis of dietary intervention studies showed that the intake of eggs (range 1–4 eggs per day, duration 5–48 weeks) increased macular pigment optical density and serum lutein concentrations by reducing AMD progression [40]. However, the small number of studies (*n* = 5), few participants in them (*n* = 296), and their moderate quality could not allow any definitive conclusions, as remarked in [40].

Results from observational studies failed to demonstrate health benefits or detrimental effects on cognitive health due to the consumption of eggs in older subjects. In fact, although eggs can contribute to the intake of nutrients that potentially have beneficial effects on cognitive health, the amount habitually consumed (one egg average daily intake) could be insufficient to lead to measurable changes in cognitive function [41,42,43]. The impacts of egg consumption have been evaluated in the context of CVD, and recent evidence concluded that increased egg consumption was not associated with risks in the general population [44], though a recent systematic review of 17 randomized controlled trials concluded that long-term consumption three or more eggs per day may lead to a higher LDL-C/HDL-C ratio and more LDL-C [45].

With respect to green vegetables, the main foods tested included: green peppers, green-leafy vegetables, and mixed foods (e.g., vegetable juices and vegetable salad). Most of them were mainly focused on carotenoids’ bioavailability, and few registered trials investigated their impacts on human health. The main topics included oxidative stress, inflammation, and obesity/adiposity both in young and older subjects. The main green vegetables studied included kale [46], broccoli [47,48,49], spinach [50], peas [51], and peppers [52]. These particular studies were focused on the effects on oxidative stress, lipid-lowering activity, satiety, and hypoglycemic activity. However, since these vegetables are also rich sources of numerous bioactives, it is difficult to differentiate the contributions of carotenoids from those of other compounds (e.g., glucosinolates, polyphenols, vitamins, and fiber).

Finally, while biofortified maize has been found to improve maternal and infant vitamin A status and reduce vitamin A deficiency, mango has been investigated for its potential effect on skin health and impact on microbiota composition. Regarding the latter, two studies have been recently registered. The first study investigated the effects of an 8-week intervention with mango (280 g/day, equivalent to a 2-cup serving) on skin health and gut microbiome in overweight and obese subjects (trial ended on December 2022). The second study evaluated the effects of a 4-week intervention with mango (85 g/day) on gut microbiota and metabolic health in postmenopausal women (trial ended on June 2023). The interaction between carotenoids and the gut has been poorly investigated, and mainly in animal models. A recent cross-sectional study reported that total carotenoid, beta-carotene, α-carotene, cryptoxanthin, and lycopene plasma concentrations were associated with higher gut bacterial diversity in men and women. This was the Multiethnic Cohort-Adiposity Phenotype Study (*n* = 1709). In addition, a higher total carotenoid concentration was associated with greater abundance of some genera relevant for microbial macronutrient metabolism [53]. Another recent small-scale cross-sectional study showed that dietary and plasma carotenoids were positively associated with alpha diversity in the fecal microbiota of pregnant women (*n* = 27). In particular, correlations with α and ß-carotene were found [54]. The effects of carotenoid supplementation on gut composition are few, and a recent review tried to summarize the main findings by concluding that the “carotenoids–gut” interaction still represents a fundamental research question that must be addressed [55].

## 5. Strengths and Limitations of the Study

This work has several noteworthy strengths and limitations. Regarding strengths, this study provides an overview of the ongoing research on carotenoids and an updated picture of the challenges to address. In addition, the use of databases such as ClinicalTrials.gov and the ISRCTN registry represents an important means for the retrieval of specific information often missing in publications (e.g., all outcome measures and characteristics of the study population). Moreover, the information reported here could be useful for industries in the context of research and innovation actions. In this regard, the realization of new products (or the improvement of existing ones) with potential nutritional/functional/nutraceutical value and improvement and/or innovation towards a more sustainable food system could represent important tasks for various industries.

Concerning limitations, it is worth mentioning that a search of registered clinical trials does not allow one to collect information related to the study results. In addition, we cannot exclude the possibility that we missed trials focused on carotenoids and health that have not been registered or have been registered in databases different from the two used in the present study.

## 6. Future-Study Directions

Numerous registered trials are focused on the absorption, distribution, and metabolism of carotenoids at different tissue levels (breast, prostate, and skin) in consideration of the different sources (e.g., supplements and foods). Some also use new formulations and/or alternative food sources (e.g., fungi, algae, new crops, and added-value products). In addition, the potential impacts/contributions of food processing (e.g., mild and/or alternative technologies and thermal treatments) and individual factors (e.g., age, sex, and genetic factors); and carotenoids’ bioaccessibility, bioavailability, and metabolism still remain important issues for which the research is trying to find answers. Regarding human health, the research is directed to providing further evidence about the roles of carotenoids in the prevention of vascular diseases, oxidative stress, and inflammation; and more so, in the prevention and treatment of eye disorders and diseases. Despite not being new, this topic remains largely debated, particularly when the effects of single compounds, supplements, and foods are compared.

The microbiota is also relevant to carotenoid studies. In fact, it is well known that the microbiota is involved in the biotransformation of numerous bioactive molecules (e.g., polyphenols and glucosinolates) whose metabolic products and activities are unknown. At the same time, the presence of these components could modify microbiotic composition, driving the selection towards bacterial species with positive impacts on host health. Data available on this topic are too preliminary to demonstrate direct/indirect involvement of carotenoids in the modulation of microbiota and vice versa.

Moreover, the increase in the aging population has led, in part, to a surge in demand for studies focused on carotenoids and age-related diseases, such those linked to cognitive decline. Here too, the research is still only beginning, but several studies are ongoing and focused on understanding the potential contributions of these bioactives to the prevention, control, and/or treatment of cognitive decline.

## 7. Conclusions

In conclusion, carotenoids have been and are still the object of numerous studies, as confirmed by the large number of registered clinical trials. There is strong interest in these compounds due to their wide range of applications. In fact, the current research on carotenoids is still focused on studying the bioavailability and metabolism of carotenoids from foods (by considering the impact of food technology) or from new formulations (e.g., food supplements). In addition, the research is trying to understand the cause–effect relationship linking carotenoids to vision and cardiovascular health in order to corroborate the previous findings. However, the heterogeneity of studies registered so far highlights the need for further “harmonized” and “coordinated” studies that have to be conducted in order to better elucidate carotenoids’ ties to human health. We especially need to focus on markers that have been less investigated or that have been considered only in the most recent studies. In this regard, new research areas include the study of the interactions between carotenoids and microbiota, and between carotenoids and cognitive function. Apart from in vitro and animal studies that should establish the exact mechanisms of action of carotenoids, well-controlled human intervention studies are advised. This will help to clarify the beneficial effects of carotenoids to establish future dietary recommendations.

## Figures and Tables

**Figure 1 nutrients-14-01191-f001:**
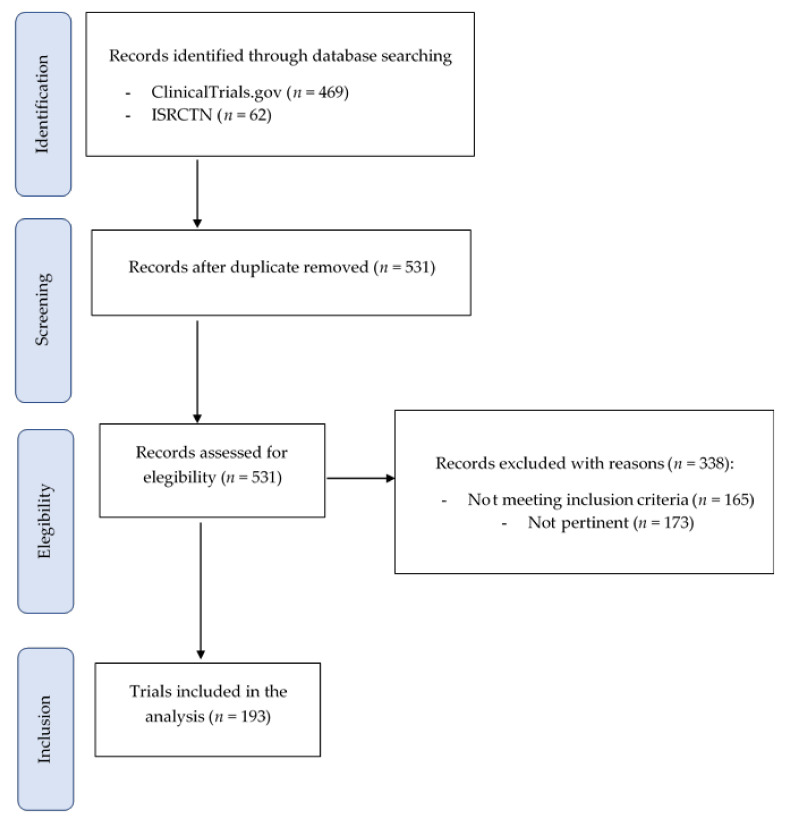
Characteristics of the included studies (*n* = 193).

**Figure 2 nutrients-14-01191-f002:**
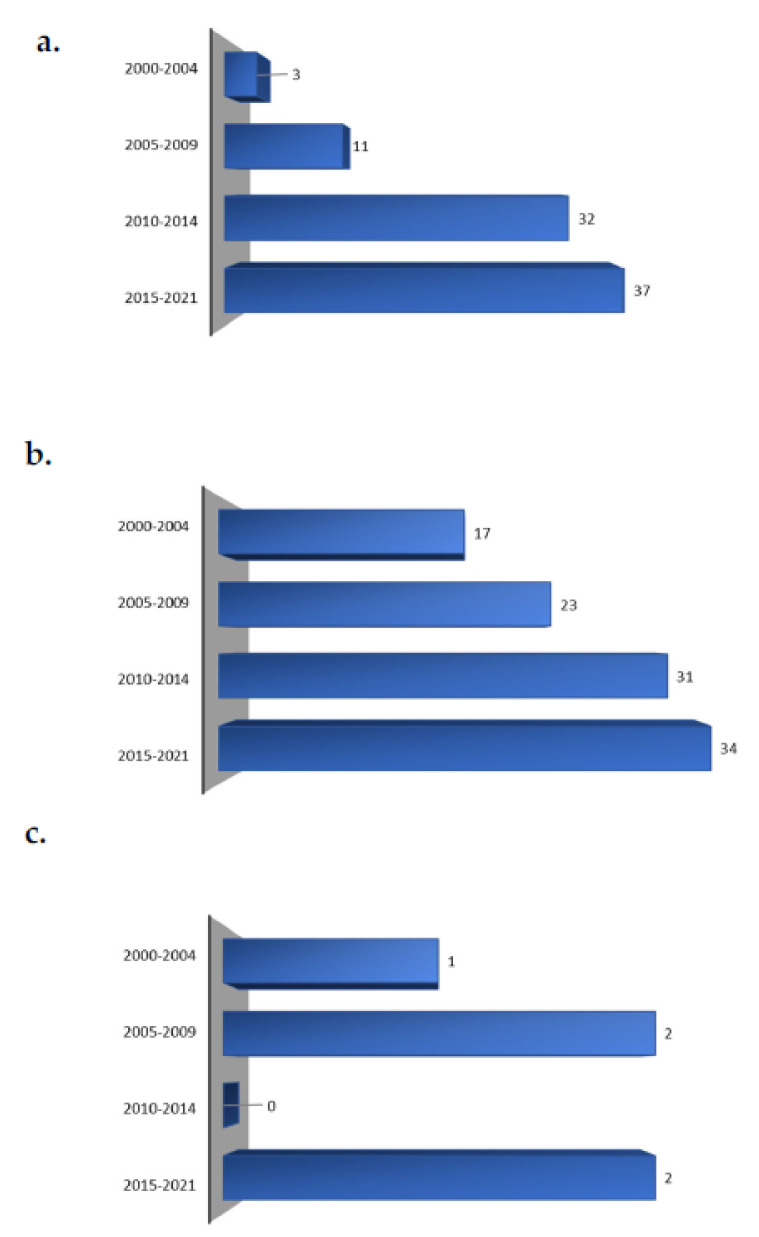
Trend of registered trials on carotenoid-rich foods (**a**), supplements (**b**) or both (**c**).

**Figure 3 nutrients-14-01191-f003:**
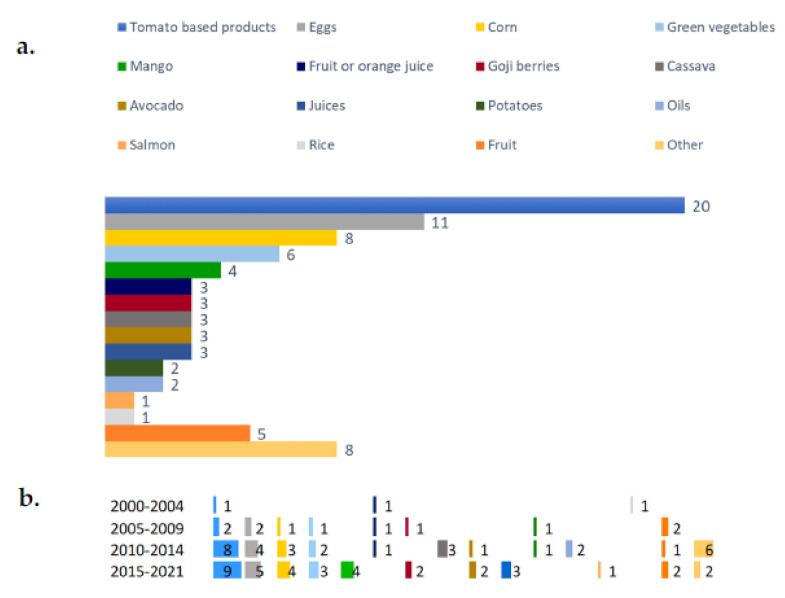
Main carotenoid-rich foods (**a**) used in registered trials and their trends (**b**).

**Figure 4 nutrients-14-01191-f004:**
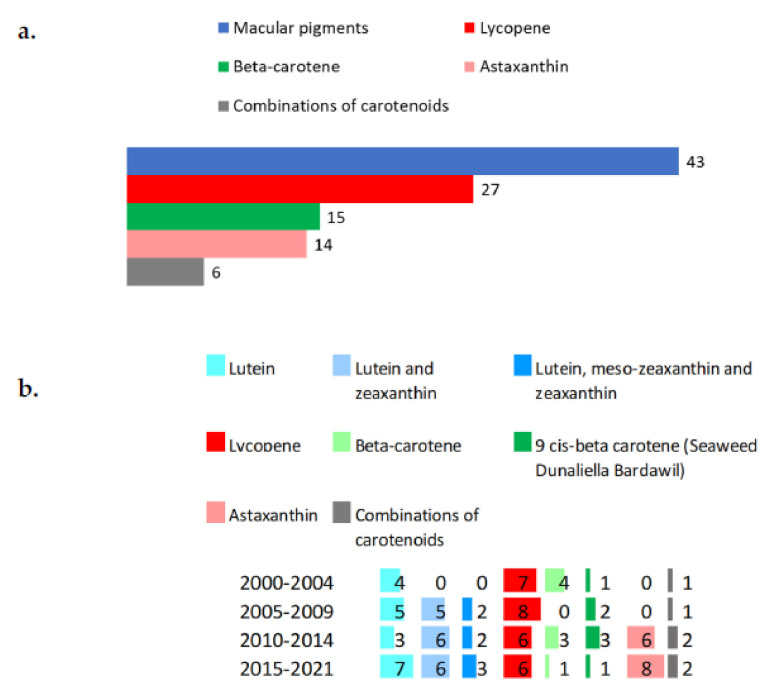
Main supplements of carotenoids (**a**) used in registered trials and their trends (**b**).

**Figure 5 nutrients-14-01191-f005:**
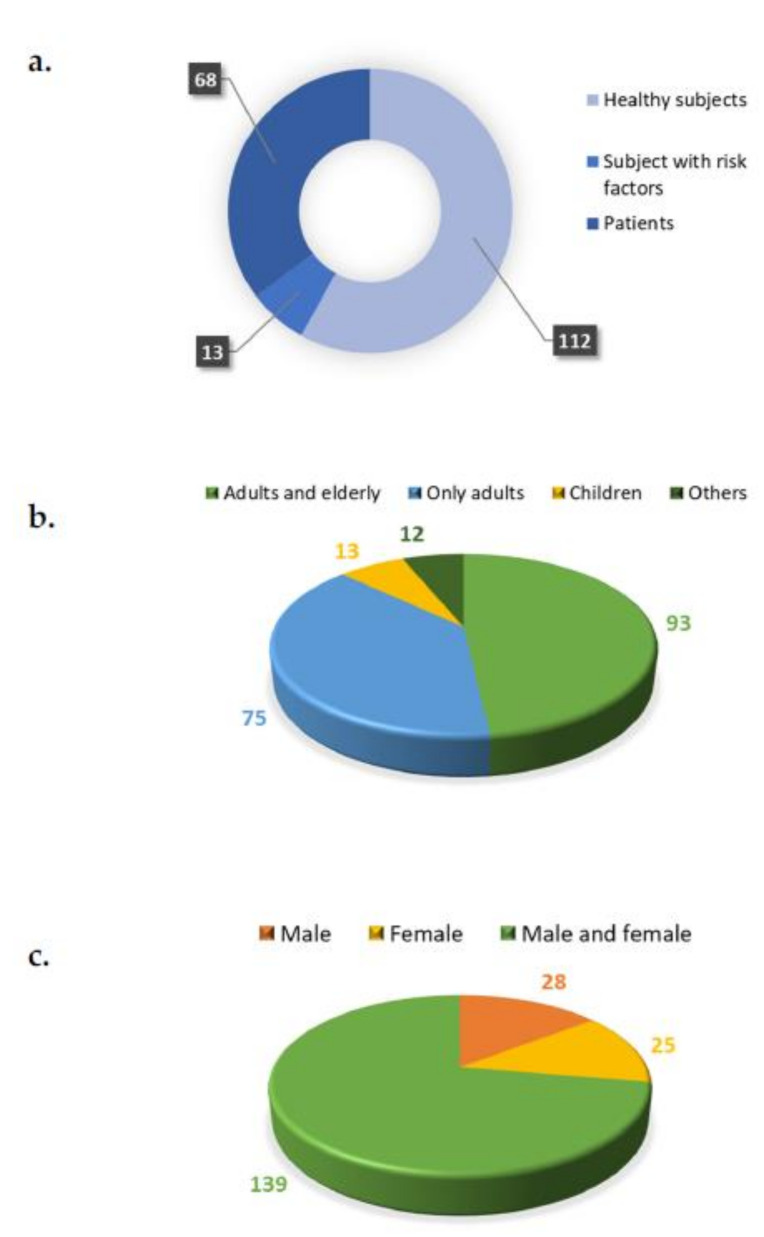
Health status (**a**), age (**b**), and sex (**c**) breakdown of subjects included in the registered trials.

**Figure 6 nutrients-14-01191-f006:**
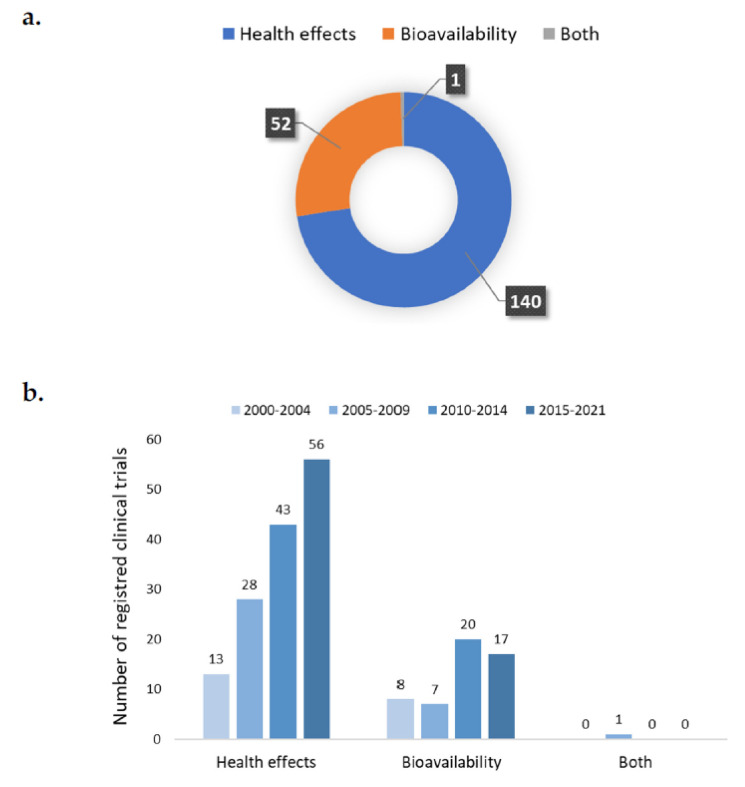
Main objectives of the registered trials (**a**) and their trends (**b**) over time.

**Figure 7 nutrients-14-01191-f007:**
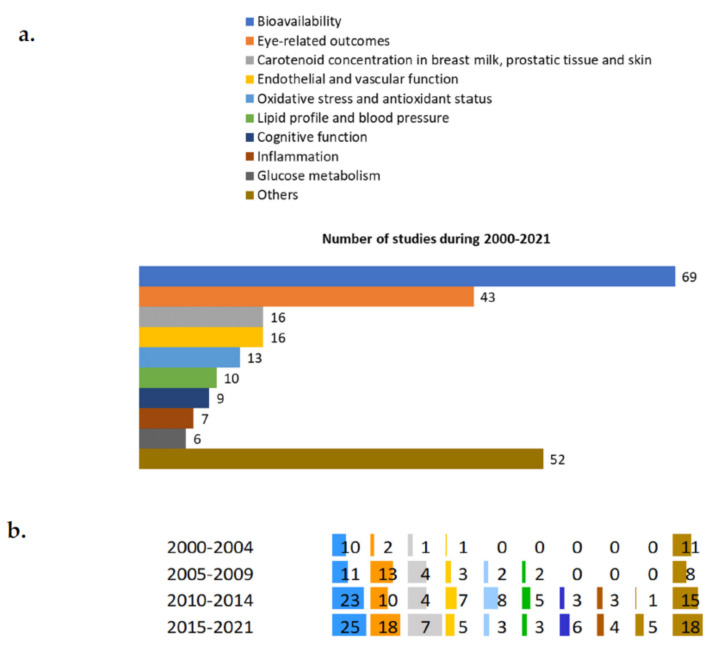
Main outcomes of the registered trials (**a**) and their trends (**b**) over time.

**Table 1 nutrients-14-01191-t001:** Population, intervention, comparison, outcome, study design (PICOS) criteria for the selection of clinical trials.

PICOS	Inclusion Criteria
Population	Healthy or diseased children, adults and/or older subjects
Intervention	Carotenoids present in food or as supplements, alone or in combination with other carotenoids
Comparison	Control group without carotenoids
Outcome	Bioavailability and/or any effect on human health
Study design	No restriction

**Table 2 nutrients-14-01191-t002:** Main characteristics of the studies included in the analysis.

	All	Food	Supplements	Food and Supplements
	(*n* = 193)	(*n* = 83)	(*n* = 105)	(*n* = 5)
*Goal*				
Human health	140	49	89	2
Bioavailability	52	33	16	3
Both	1	1	0	0
*Duration*				
Acute	33	20	11	2
Chronic	158	61	94	3
Both	2	2	0	0
*Subjects*				
Healthy	112	65	43	4
With risk factors	13	5	8	0
Patients	68	13	54	1
*Primary outcome*				
Bioavailability	69	42	23	4
Eye-related outcomes	43	7	35	1
Carotenoid concentration in breast milk, prostatic tissue and skin	16	9	7	0
Endothelial and vascular function	16	9	7	0
Oxidative stress and antioxidant status	13	4	9	0
Lipid profile and blood pressure	10	7	3	0
Cognitive function	9	6	3	0
Inflammation	7	6	1	0
Glucose metabolism	6	4	2	0
Others	52	16	36	0
*Country*				
United States of America	74	35	37	2
Israel	13	0	13	0
United Kingdom	12	4	8	0
Italy	8	3	5	0
The Netherlands	7	3	3	1
Ireland	6	1	5	0
France	6	4	2	0
China	5	1	4	0
Not provided	14	7	6	1
Others	48	25	22	1

## Data Availability

Not applicable.

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
