# Peer review of "What Is the Current Direction of the Research on Carotenoids and Human Health? An Overview of Registered Clinical Trials"

_nutrients, 2022, doi:10.3390/nu14061191_

Round 1
Reviewer 1 Report
The authors carried out a limited scoping review of registered clinical trials focused on carotenoids, bioavailability and human health outcomes. While this is an unique approach to review recently funded work, this will not provide a comprehensive overview of findings of studies for the reader. The authors cannot provide a critical review of relationships of carotenoids from foods and/or supplements to health as only registered trials are included. There are some other concerns.
- There are a number of spelling errors, incorrect use of phrases, tense, and the manuscript is very wordy - especially the abstract and introduction sections. English editing is needed.
- The authors should provide in Supplemental Information a listing of references to all 193 included trials.
- Under 3.2., what is "orange sweet"?
- In the Discussion section, a reference should be provided for the estimate of USD 2.0 billion dollars growth.
- The Conclusion section seems unrelated to the manuscript. Rewrite.
Author Response
The authors carried out a limited scoping review of registered clinical trials focused on carotenoids, bioavailability and human health outcomes. While this is an unique approach to review recently funded work, this will not provide a comprehensive overview of findings of studies for the reader. The authors cannot provide a critical review of relationships of carotenoids from foods and/or supplements to health as only registered trials are included. There are some other concerns.
R: We thank the Reviewer for noting that our study used a unique approach for by collecting information from registered clinical trials. We agree that we cannot provide a critical review of the relationship between carotenoid and human health (this was already reported in the section “strength and limitation”). We have modified according to your comments the aim of the manuscript that remains to provide an overview of the current ongoing research on carotenoids. In this regard, the relationship between carotenoids and health effects has been the object of recent scientific papers and this work did not want to be a duplicate. However, we want to underline that together with registered clinical trials we have also reported, when available, results/conclusions from systematic reviews and meta-analysis of RCT that partially overcame the lack of findings.
Overall, we do believe, as also underlined by reviewer 2, that this study may represent an important instrument and a valid strategy both for industries and researchers working on carotenoids to collect information on the current research topics and drive future studies and efforts where required.
There are a number of spelling errors, incorrect use of phrases, tense, and the manuscript is very wordy - especially the abstract and introduction sections. English editing is needed.
R: We have carefully revised English language as requested by the reviewer, by using the MDPI English Editing Service
The authors should provide in Supplemental Information a listing of references to all 193 included trials.
R: We have included a list of the trials as supplementary material.
Under 3.2., what is "orange sweet"?
R: We thank the Reviewer for noting this typo. We referred to sweet orange and not to orange sweet, we have revised it.
In the Discussion section, a reference should be provided for the estimate of USD 2.0 billion dollars growth.
R: We thank the Reviewer for this suggestion. The data comes from BCC research and we have added the reference in the Discussion, as suggested.
The Conclusion section seems unrelated to the manuscript. Rewrite.
R: We thank the reviewer for this comment. Accordingly, we have revised the Conclusion section.

Reviewer 2 Report
This is a novel article, correctly designed which gives an overview of current direction of the research on carotenoids in human health. It is well planned and designed and data are presented appropriately which makes the article easy to read and understand. This article may attract the attention of researchers working in the area of carotenoids
Author Response
This is a novel article, correctly designed which gives an overview of current direction of the research on carotenoids in human health. It is well planned and designed and data are presented appropriately which makes the article easy to read and understand. This article may attract the attention of researchers working in the area of carotenoids
R: We thank a lot the Reviewer for the congratulations.
